# Structural basis for recognition of the malaria vaccine candidate Pfs48/45 by a transmission blocking antibody

Frank Lennartz[1], Florian Brod[2], Rebecca Dabbs[2], Kazutoyo Miura[3], David Mekhaiel[2], Arianna Marini[2], Matthijs M. Jore[4], Max M. Søgaard[5], Thomas Jørgensen[5], Willem A. de Jongh[5], Robert W. Sauerwein[4], Carole A. Long[3], Sumi Biswas [2] & Matthew K. Higgins [1]

The quest to develop an effective malaria vaccine remains a major priority in the fight against global infectious disease. An approach with great potential is a transmission-blocking vaccine which induces antibodies that prevent establishment of a productive infection in mosquitos that feed on infected humans, thereby stopping the transmission cycle. One of the most promising targets for such a vaccine is the gamete surface protein, Pfs48/45. Here we establish a system for production of full-length Pfs48/45 and use this to raise a panel of monoclonal antibodies. We map the binding regions of these antibodies on Pfs48/45 and correlate the location of their epitopes with their transmission-blocking activity. Finally, we present the structure of the C-terminal domain of Pfs48/45 bound to the most potent transmission-blocking antibody, and provide key molecular information for future structure-guided immunogen design.

[1] Department of Biochemistry, University of Oxford, South Parks Road, Oxford OX1 3QU, UK. [2] Jenner Institute, University of Oxford, Old Road Campus Research Building, Roosevelt Drive, Oxford OX3 7DQ, UK. [3] Malaria Immunology Section, Laboratory of Malaria and Vector Research, National Institute of Allergy and Infectious Diseases, NIH, Rockville 20852 MD, USA. [4] Department of Medical Microbiology, Radboud University Medical Centre, Nijmegen PO Box 9101, 6500 HB, The Netherlands. [5] ExpreS2ion Biotechnologies, SCION-DTU Science Park, Agern Alle 1, DK-2970 Horsholm, Denmark. These authors contributed equally: Frank Lennartz, Florian Brod. Correspondence and requests for materials should be addressed to S.B. (email: sumi.biswas@ndm.ox.ac.uk) or to M.K.H. (email: matthew.higgins@bioch.ox.ac.uk)

Malaria is one of the most devastating diseases to affect humanity, causing hundreds of millions of cases and around half a million deaths each year[1]. The development of a successful malaria vaccine is therefore pressing. However, the malaria parasite is an ancient organism that has been co-evolving with humans for millennia, and generation of a vaccine has proved a major challenge. In particular, the life cycle of the parasite is complex and involves multiple developmental stages in both the human host and the mosquito vector. Additionally, parasites surface proteins frequently adapt rapidly to avoid immune detection through antigenic variation. With many individual immunogens having already been tested in vaccine trials with varied success, it is commonly acknowledged that an effective vaccine will contain multiple components that represent multiple stages of the parasite life cycle[2]. Immunogens which raise antibodies that interrupt the life cycle at the sexual stage will prevent transmission of malaria from infected human to mosquito and are potential components of such a vaccine.

The sexual cycle of the *Plasmodium* life cycle occurs when male and female gametocytes are ingested as part of a blood meal, leading to their differentiation into male and female gametes within the midgut of an infected mosquito. The gametes fuse to form zygotes, which then develop into oocysts, allowing the parasite life cycle to continue as emerging sporozoites relocate into the mosquito salivary glands, positioned to infect other humans. Several proteins are found on the surfaces of both gametocytes and gametes and play critical roles in this sexual event. In particular, Pfs48/45 and Pfs230 form a Glycosylphosphatidylinositol (GPI)-anchored complex on the gametocyte surface and are required for *Plasmodium* gamete fusion[3–5].

A number of factors converge to suggest Pfs48/45 as a leading candidate for inclusion in a transmission-blocking vaccine[6]. *Plasmodium falciparum* parasites that do not express Pfs48/45 are severely impaired in their ability to form ookinetes in mosquitoes[5]. Studies using the rodent malaria species *Plasmodium berghei* suggest that this is due to an inability of gametes lacking the Pfs48/45 orthologue Pbs48/45 to penetrate female gametes and to proceed to form zygotes[5]. Indeed, sera from animals immunised with Pfs48/45 contain antibodies that, when present in a parasite-infected blood meal, block the sexual and sporogonic development of the parasite within the infected mosquito[7–13]. In addition, unlike other transmission-blocking vaccine candidates, Pfs48/45 and Pfs230 are expressed in gametocytes found in human blood and the presence of antibodies that target Pfs48/45 in individuals from malaria-endemic regions correlates with the transmission-blocking activity of their sera[13–19]. Recently, it has been demonstrated that specific antibodies in endemic sera against Pfs48/45 can functionally block transmission of *Plasmodium falciparum* in infected mosquitoes in a standard membrane-feeding assay (SMFA)[20]. Individuals immunised with Pfs48/45 could therefore experience immune boosting through natural low-level infection. Finally, unlike many important *Plasmodium* surface antigens, sequence diversity of Pfs48/45 is low across strains of *Plasmodium falciparum*[19,21]. Altogether, this suggests that a vaccine immunogen based on Pfs48/45 will generate antibodies that target a conserved and essential component of the parasite life cycle and will prevent further transmission to uninfected individuals.

To understand the targets of Pfs48/45 reactive antibodies, a number of studies have generated and characterised monoclonal antibodies. Antibodies cloned from immunised mice and rats can block oocyst development in mosquitoes at close to 100% in an SMFA[8,13,22]. These have been sorted into five competition groups and members of four of these groups have transmission-blocking activity[22–24]. Pfs48/45 contains two 6-cys domains, a domain type found among *Plasmodium* proteins expressed throughout

different life cycle stages[25], which in Pfs48/45 are separated by a 4-cys linker domain[26]. As Pfs48/45 had previously been difficult to express, assessment of Pfs48/45 as a transmission-blocking antigen has focused on truncation variants, containing the central and the C-terminal 6-Cys domain (Pfs48/45-10C) or the C-terminal 6-Cys domain alone (Pfs48/45-6C). The C-terminal 6-cys domain is described as 6C, and a construct which lacks just the N-terminal 6-cys domain is known as 10C[10] (Fig. 1a). The most effective monoclonal antibodies characterised to date, 85RF45.1[13] and 32F3[8] bind to 6C, highlighting the C-terminal domain of Pfs48/45 as a promising vaccine candidate. However, not all inhibitory antibodies bind to 6C[6,24,27], and our understanding of the interplay of different inhibitory epitopes, as well as the development of Pfs48/45 as vaccine candidate, have been hampered by lack of a system to generate correctly folded full-length protein. Furthermore, the absence of a structure of Pfs48/45 bound to transmission-blocking antibodies has prevented rational structure-guided immunogen design.

We have therefore developed an expression system for full-length Pfs48/45 that allows for the production of milligram quantities of correctly folded Pfs48/45. Using this protein, we have raised and characterised a panel of monoclonal antibodies, demonstrating that transmission-blocking antibodies bind to the central and the C-terminal domain of Pfs48/45. Finally, we have determined the crystal structure of Pfs48/45 bound to the most potent transmission-blocking antibody and show that the epitope targeted by this antibody is highly conserved among all characterised *Plasmodium falciparum* isolates.

## Results

**Insect cell produced Pfs48/45 elicits inhibitory antibodies**. A major obstacle for development of Pfs48/45 as a transmission-blocking vaccine has been the lack of an expression system that produces large quantities of correctly folded full-length protein. The most effective solution to date has been to express truncated forms of Pfs48/45. Constructs containing either the C-terminal domain (6C) or the central and C-terminal domains (10C) (Fig. 1a) can be expressed in *Lactococcus lactis* when fused to the asexual blood-stage malaria antigen GLURP[28], but this results in problems associated with a large fusion partner. Therefore, non-virally transfected *Drosophila* Schneider-2 cells were used to produce full-length Pfs48/45 lacking its GPI-anchor but with glycosylation sites intact (Fig. 1b) at 2 mg per litre. This protein was pure and was recognised by antibodies that bind conformational epitopes in the central (85RF45.3) and C-terminal (85RF45.1 and 32F3) domains[13] (Fig. 1b), suggesting that it has adopted the correct fold. This was further confirmed by circular dichroism spectroscopy, showing that recombinant Pfs48/45 predominantly consist of β-sheets (Supplementary Figure 1A), as expected due to the presence of 6-cys domains[25].

To assess the ability of this Pfs48/45 to induce functionally active transmission-blocking antibodies, CD1 mice were immunised. These outbred (CD1) mice were used to maximise the breadth of the induced antibody response. Serum antibody titres were determined by end point ELISA, revealing an immune response that was boosted at each of the three time points (Fig. 1c). We confirmed the functionality of the induced antibodies by SMFA, in which purified IgG from pooled serum samples from mice immunised with Pfs48/45 reduced the oocyst intensity of mosquitoes fed with gametocytes by 96% (95% CI = 87.7–98.6; $p = 0.001$) when compared with IgG from pooled serum samples from mice immunised with OVA (Fig. 1d and Supplementary Figure 1B). This gave comparable levels of transmission-blocking activity to a positive control, the Pfs25-reactive monoclonal antibody 4B7. These results were confirmed

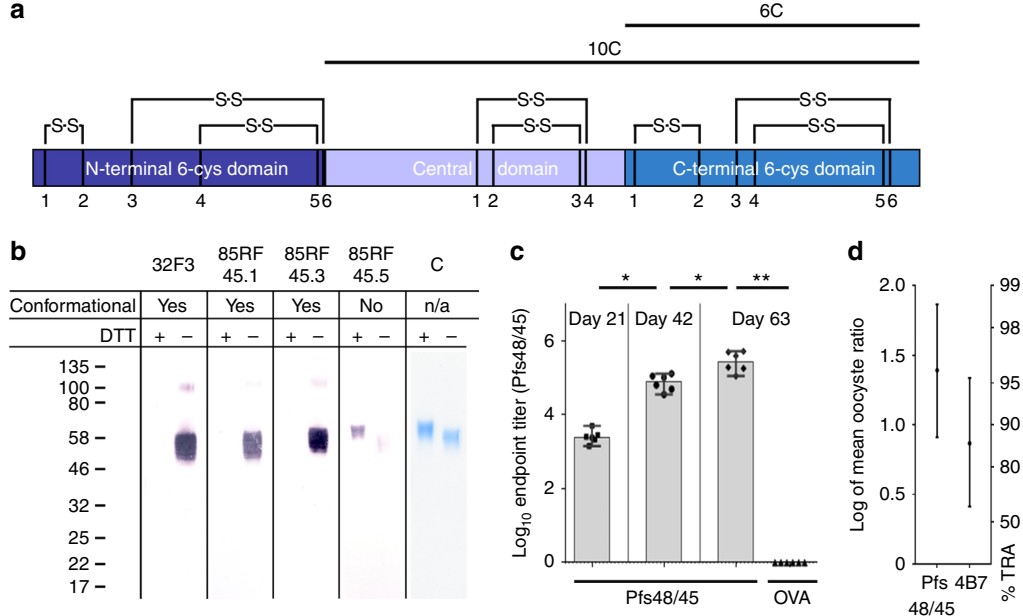

**Fig. 1** Full-length Pfs48/45 can be expressed in native conformation by S2 cells. **a** Domain structure of Pfs48/45. Length of bars is proportional to the number of amino acids in each domain. Disulphide bonds are drawn based on sequence homology to other 6-Cys domain proteins and numbered for each individual domain. **b** Western blots and Coomassie gel of full-length Pfs48/45. The protein was run on 4–12% Bis-Tris polyacrylamide gels in the presence and absence of dithiothreitol (DTT) and stained with Coomassie brilliant blue (**c**) or blotted on nitrocellulose membranes and detected with Pfs48/45-specific monoclonal antibodies (32F3, 85RF45.1, 85RF45.3 and 85RF45.5, at a concentration of 1 μg/ml). **c** Anti-Pfs48/45 IgG end point titres in mouse serum after immunisation with Pfs48/45 or chicken ovalbumin (OVA) ($n = 6$). Data points show end point titres in individual mice determined by Pfs48/45 end point ELISA, bars show median, error bars show 95% CI. Anti-Pfs48/45 IgG titres after consecutive immunisations were compared by Wilcoxon signed-rank test, day 63 anti-Pfs48/45 IgG titres induced by immunisation with Pfs48/45-FL and OVA, respectively, were compared by Mann–Whitney test, $*p < 0.05$, $**p < 0.005$. **d** Transmission-reducing activity of purified IgG from mice immunised with Pfs48/45 was determined relative to purified IgG from mice immunised with OVA at 750 μg/ml. Transmission-blocking mAb 4B7 was used as a positive control at a concentration of 94 μg/ml. Data points show transmission-reducing activity (TRA) calculated from the oocyst counts in 20 mosquito midguts, error bars show 95% CI

in an independent mouse experiment, which furthermore revealed that complement was not required for anti-Pfs48/45-mediated transmission-reducing activity (Supplementary Figure 1C and D). Therefore, full-length Pfs48/45 expressed in insect cells contains epitopes bound by key monoclonal antibodies and is able to potently raise a transmission-blocking immune response.

**Monoclonal antibodies raised by immunisation with Pfs48/45.** With full-length Pfs48/45 available for the first time, we next raised and screened a panel of monoclonal antibodies, to identify which regions of this protein contain epitopes with the potential to generate transmission-blocking responses. Hybridoma colonies were selected from splenocytes of Balb/c mice immunised with Pfs48/45. Balb/c mice were chosen as previous studies had shown that mAbs with high transmission-reducing activity can be raised in this mouse strain[8]. Of these, 16 produced antibodies that bound Pfs48/45 at levels above background reactivity in ELISA.

These antibodies were further analysed by cross-competition ELISA, to find groups that share overlapping epitopes (Fig. 2a, b and Supplementary Figure 2). We identified three groups, the largest of which contains eight mAbs (group A), while the other two groups (B and C) each contained two mAbs. The four remaining antibodies did not belong to any competition group. The interactions between these mAbs were complex. For example, 9D1 was only efficiently blocked by one member of competition group A (1F10), while 9A6 was efficiently blocked by all mAbs in group A, but did not itself block them. Finally, 7A6 was blocked by 10D8 (from group A), and itself blocked mAbs of group C as well as 9A6. The only mAb to not show any competition with any

other mAb was 10F10. This complex network of interactions suggests that a variety of overlapping epitopes are present within Pfs48/45. No mAb from this panel blocked binding of the previously identified transmission-blocking mAbs 85RF45.1 and 32F3.

The ability of the generated mAbs to bind native parasite epitopes was assessed by indirect fluorescence assay. Out of the 16 mAbs, 13 bound *Plasmodium falciparum* gametes air dried onto glass microscopy slides (Supplementary Figure 3), suggesting that the majority of the epitopes are exposed on the gamete surface. The functional activity of the mAbs was next assessed by SMFA (Fig. 2c and Supplementary Figure 4A–C). Four mAbs (6A10, 3G3, 10D8 and 1F10), all of which are members of competition group A, reduced transmission. At a concentration of 375 μg/ml, the transmission-reducing activity from two feeds was calculated as 74.5% (95% CI = 45.7–88.1%; $p = 0.001$) for 6A10, 59.1% (95% CI = 14.1–80.8%; $p = 0.019$) for 3G3, 55.9% (95% CI = 6.9–78.7%; $p = 0.031$) for 10D8 and 74.5% (95% CI = 46.9–88.3%; $p = 0.001$) for 1F10. There was no connection between transmission-reducing activity and the isotype or Pfs48/45 end point titre of the mAbs.

Our new transmission-blocking antibodies require significantly higher concentrations to achieve effective blocking activity when compared to the levels previously cited for antibody 85RF45.1, which is completely effective at 12.5 μg/ml[13]. To further evaluate these differences, the four blocking mAbs of our panel were compared side by side with the well-characterised transmission-blocking mAbs 85RF45.1 and 32F3 in SMFA at different concentrations (Fig. 2d and Supplementary Figure 4D). In this assay, only 1F10 showed significant TRA, and only at the highest-

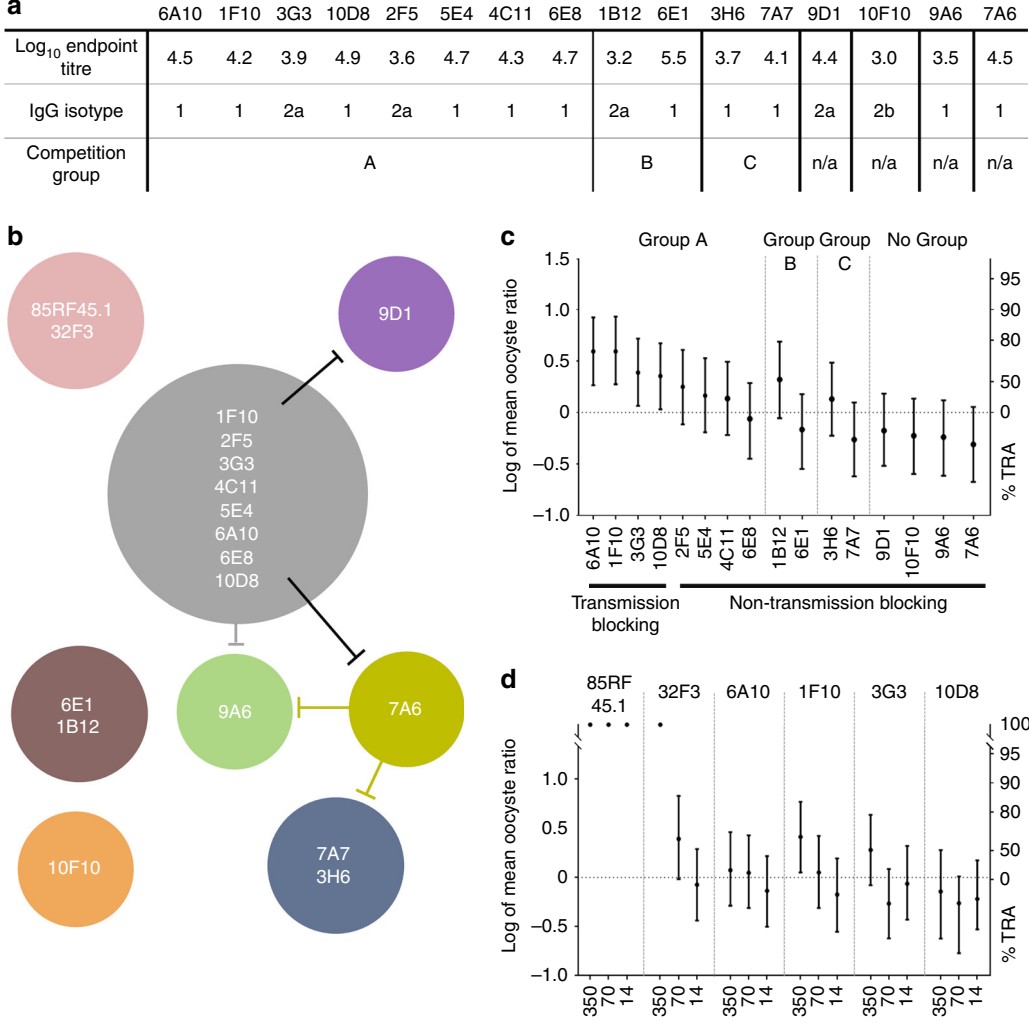

**Fig. 2** Full-length Pfs48/45 induces functional polyclonal and monoclonal antibodies in mice. **a**. A panel of Pfs48/45-specific mAbs was raised from splenocytes of mice immunised with full-length Pfs48/45. End point titres were determined by Pfs48/45 end point ELISA. The IgG isotypes of the individual mAbs are indicated. The mAbs were grouped by cross-competition ELISA. **b** Schematic depiction of competition groups in the mAb panel and 85RF45.1 and 32F3. Lines indicate cross-competition between whole groups or individual mAbs and other competition groups. **c** Transmission-reducing activity of the anti-Pfs48/45 mAb panel at a concentration of 375 μg/ml was determined relative to normal mouse antibody. Data points show %TRA calculated from the oocyst counts in 20 mosquito midguts for non-transmission-blocking mAbs and 40 mosquito midguts from two independent feeds for transmission-blocking mAbs, error bars show 95% CI. **d** Transmission-reducing activity of selected anti-Pfs48/45 mAbs was determined relative to normal mouse IgG at 350, 70 and 14 μg/ml. Data points show transmission-reducing activity calculated from the oocyst counts in 20 mosquito midguts, error bars show 95% CI

tested concentration of 350 μg/ml (%TRA = 61.0%; 95% CI = 11.0–82.8%; p = 0.032). Similarly, 32F3 showed significant TRA only at 350 μg/ml (%TRA = 99.8%; 95% CI = 98.9–100%; p = 0.032), while 85RF45.1 showed 100% TRA at all tested concentrations (p < 0.001 at all concentrations). Our next aim was therefore to map the binding region of these antibodies onto the three domains of Pfs48/45, to understand if transmission blocking is specifically associated with epitopes in one of these domains.

**Identifying the domains targeted by inhibitory antibodies.** The availability of a comprehensive panel of both new and previously characterised antibodies provided the opportunity to explore the molecular determinants of transmission blocking in more detail. For this, we mapped the binding region of these antibodies onto the three domains of Pfs48/45. In addition to full-length Pfs48/45, we produced the C-terminal 6-cys domain (6C) and the 10C

construct containing the central and C-terminal domains (Fig. 1a) in Schneider-2 cells and confirmed their correct fold by CD spectroscopy and western blot (Supplementary Figure 5A and B). We then used surface plasmon resonance and dot blots to study binding of all three constructs to our extended panel of antibodies (Fig. 3a and Supplementary Figure 5C).

Overall, the antibodies bound with varying strength to the different Pfs48/45 constructs, independent of their blocking or non-blocking activity (Fig. 3a). In particular, 9A6 showed very weak binding to Pfs48/45, while 3H6 and 7A7 bound Pfs48/45 with quick dissociation kinetics, indicating a very short-lived interaction. This suggests that the inability of these mAbs to detect gametes in the indirect immunofluorescence assay (Supplementary Figure 3) could be due to their binding affinity and kinetics rather than due to inaccessibility or non-native conformation of their epitopes. With regard to the individual domains bound by the antibodies, we found that members of antibody competition group B (1B12 and 6E1, Fig. 2b), both of

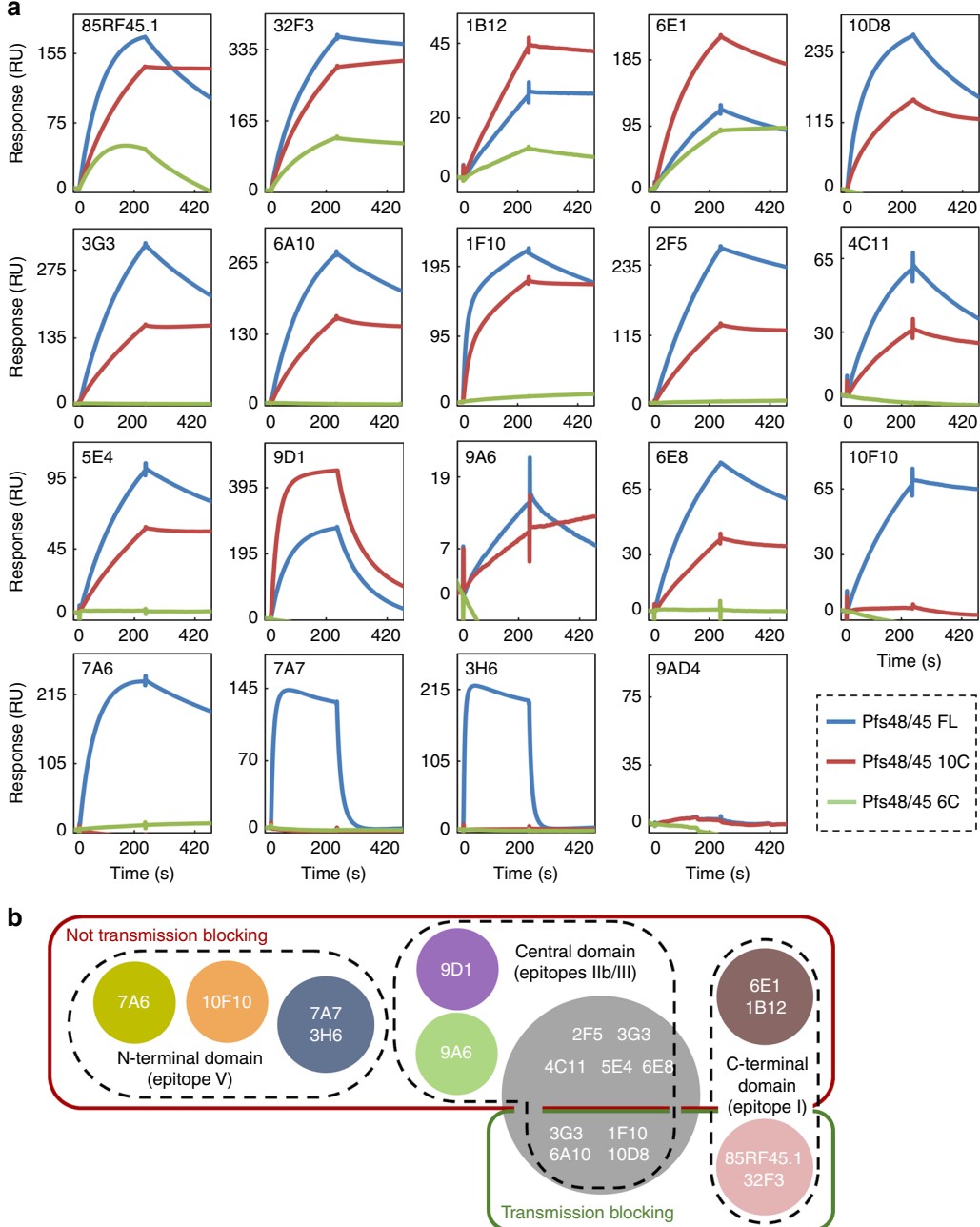

**Fig. 3** Mapping of anti-Pfs48/45 antibodies binding to different Pfs48/45 subdomains. **a** The indicated mAbs were immobilised on a Protein A/G chip at fixed concentration. Pfs48/45 FL (blue lines), Pfs48/45-10C (red lines) or Pfs48/45-6C (green lines) were then injected over the chip surface at fixed concentration. 9AD4 is a control antibody reactive against PfRH5. **b** Summary of epitope mapping experiments. Circles indicate the different competition groups among the mAbs, as shown in Fig. 2. mAbs are further grouped into transmission-blocking and non-transmission-blocking mAbs by green and red outlines, respectively. Black dashed outlines show which mAbs bind to the N-terminal, central (10C) or C-terminal (6C) domain. Previously defined epitopes I–V[6], for anti-Pfs48/45 antibodies, are indicated

which are non-inhibitory, bind to all three protein constructs, suggesting that their epitopes are largely contained within the C-terminal domain of Pfs48/45. We observed a similar binding profile for 32F3 and 85RF45.1, confirming previous results[10,13]. Therefore, while the most potent transmission-blocking antibodies currently available bind to the C-terminal domain of Pfs48/45, not all antibodies that target this domain have transmission-blocking activity.

In contrast, all antibodies within group A as well as 9D1 and 9A6, bind to 10C but not to 6C, showing that a substantial part of

their epitope lies in the central domain of Pfs48/45. This includes antibodies 6A10, 1F10, 3G3 and 10D8, which show transmission-blocking activity. Finally, the antibodies of group C (3H6 and 7A7), together with 10F10 and 7A6, bind to full-length Pfs48/45, but not to 6C or 10C, showing their epitopes to be largely contained within the N-terminal domain. These show no transmission-blocking activity.

These studies confirm the C-terminal domain of Pfs48/45 as the target for the most effective transmission-blocking antibody available to date. However, they further highlight that antibodies

**Table 1 Data collection and refinement statistics**

| | Pfs48/45-6C–85RF45.1 Fab |
|---|---|
| *Data collection* | |
| Space group | $P2\,2_1 2_1$ |
| Cell dimensions | |
| a, b, c (Å) | 59.69, 165.39, 189.66 |
| α, β, γ (°) | 90.00, 90.00, 90.00 |
| Resolution (Å) | 41.98–3.23 (3.29–3.23) |
| $R_{merge}$ | 0.38 (2.08) |
| $R_{pim}$ | 0.16 (0.86) |
| I/σI | 5.4 (1.3) |
| CC1/2 | 0.98 (0.56) |
| Completeness (%) | 100 (100) |
| Redundancy | 6.5 (6.8) |
| *Refinement* | |
| Resolution (Å) | 3.23 |
| No. unique reflections | 30,979 (3059) |
| $R_{work}/R_{free}$ | 26.10/28.21 |
| No. atoms | |
| Protein | 8376 |
| Ligand/ion | — |
| Water | — |
| B-factors | |
| Protein | 79.49 |
| Ligand/ion | — |
| Water | — |
| R.m.s. deviations | |
| Bond lengths (Å) | 0.005 |
| Bond angles (°) | 0.85 |

Values in parentheses are for highest-resolution shell

binding to the central domain of Pfs48/45 also display transmission-blocking potential and demonstrate that not all epitopes within the C-terminal domain elicit effective antibody responses (Fig. 3b). Therefore, to develop Pfs48/45 into an effective vaccine candidate, a more detailed molecular understanding of critical epitopes targeted by transmission-blocking antibodies is necessary.

**Structure of the C-terminal Pfs48/45 domain with 85RF45.1**. Since the C-terminal 6-cys domain of Pfs48/45 is the target of the most potent transmission-blocking antibodies, we used structural studies to understand the architecture of this domain as well as the key epitope targeted by antibody 85RF45.1. For this, we set up crystallisation trials for a complex of a Fab fragment from 85RF45.1 bound to Pfs48/45-6C, which yielded crystals diffracting to a resolution of 3.2 Å (Table 1). We solved the structure by molecular replacement using a poly-alanine model of a sequence-related Fab fragment[29] as search model and found two copies of the 85RF45.1 Fab–Pfs48/45-6C complex in the asymmetric unit of the $P2\,2_1\,2_1$ unit cell (Fig. 4a).

Pfs48/45-6C adopts a typical β-sandwich fold, which is further characterised by three disulphide bridges, two of which connect β-strands while the third stabilises the packing of a loop connecting β-sheets 3 and 4 against the core of the domain (Fig. 4b). While most of the loops connecting the sheets of the β-sandwich are well resolved, a large loop that bridges β-sheets β4 and β5 in Pfs48/45-6C is disordered in our structure, indicating flexibility (Fig. 4b). The overall architecture and disulphide pattern of Pfs48/45-6C is highly similar to the fold of the C-terminal 6-cys domains (D2) from the two other structurally characterised *Plasmodium* 6-cys proteins, Pf12 and Pf41[25,30–32], both of which are expressed on the merozoite surface. Indeed, Pfs48/45-6C overlays with Pf41-D2 and Pf12-D2 with low root

mean square deviations of 1.27 Å and 1.65 Å (Fig. 4c) at sequence similarities of 31.6% and 34.7%, respectively. This structural similarity between proteins expressed at different stages of the parasite life cycle further highlights the 6-cys domain as a basic structural building block that can be adopted for a variety of different functions in other *Plasmodium* 6-cys proteins.

**85RF45.1 targets a conserved epitope on Pfs48/45**. The crystal structure also reveals how an effective transmission-blocking antibody binds to Pfs48/45. Antibody 85RF45.1 approaches Pfs48/45-6C from the opposite side relative to the GPI-anchored C terminus of Pfs48/45 (Fig. 4a and Supplementary Figure 6) and the interaction is mediated by five of its CDR loops (CDR H1–3, CDR L1 and L2) (Figs. 4a, 5). These CDR loops form a positively charged groove, part of which binds to a corresponding negative patch on the apical tip of Pfs48/45. Indeed, the majority of the interactions between 85RF45.1 and Pfs48/45-6C are mediated by hydrogen bonds (Fig. 5a, Supplementary Figure 7 and Supplementary Table 1), from both the heavy chain and light chain CDRs. Furthermore, CDR H2 and H3 each present hydrophobic residues (I54 and M102), which form van-der-Waals interactions with hydrophobic patches on the surface of Pfs48/45-6C (Fig. 5b, Supplementary Figure 7 and Supplementary Table 1). Together, these interactions lead to a total buried surface area of 970 Å$^2$, defining an elongated epitope on Pfs48/45-6C (Fig. 5c).

We next investigated the conservation of the 85RF45.1 epitope among Pfs48/45 from different *Plasmodium falciparum* isolates. We analysed the frequency of single-nucleotide polymorphisms that lead to amino acid changes over more than 2400 Pfs48/45 sequences from the Pf3k database. The sequence variation observed over the whole of Pfs48/45 was minor, with only two substitutions at a frequency >10% (Supplementary Figure 8). When mapped onto the surface of Pfs48/45-6C, there are no polymorphisms with significant coverage within the epitope. The only two polymorphisms that lie within the epitope occur with low frequency (I349V, 0.0092% and K416N, 0.053%) and result in substitution by chemically equivalent amino acids (Fig. 5c and Supplementary Figure 8), which would not be expected to affect binding of 85RF45.1. Therefore, 85RF45.1 targets an epitope that is highly conserved among Pfs48/45 from all characterised *Plasmodium falciparum* isolates.

## Discussion

A renewed focus on the goal of malaria eradication has brought the concept of a transmission-blocking vaccine back to prominence. Such a vaccine must efficiently target key stages of the sexual reproduction cycle of the parasite in order to prevent its transmission among a susceptible population and thereby reduce the number of malaria cases. Here, we have focused on the leading candidate for a malaria transmission-blocking vaccine, Pfs48/45. A major challenge for developing a Pfs48/45-based vaccine has been the lack of a suitable expression system for full-length protein (reviewed in ref. [6]). The insect cell-based expression system developed here overcomes this obstacle and allows the production of milligram quantities of correctly folded Pfs48/45, as well as its subunits 10C and 6C, without the need for additional fusion partners or carrier proteins. As well as generating material which can be directly included in future vaccines, the availability of full-length Pfs48/45 has allowed us to raise and analyse monoclonal antibodies and will, in the future, allow the study of naturally acquired transmission-blocking anti-Pfs48/45 antibodies from malaria-exposed individuals and the characterisation of their epitopes.

Our immunisation studies, performed with full-length Pfs48/45, show that transmission-blocking antibodies not only target its

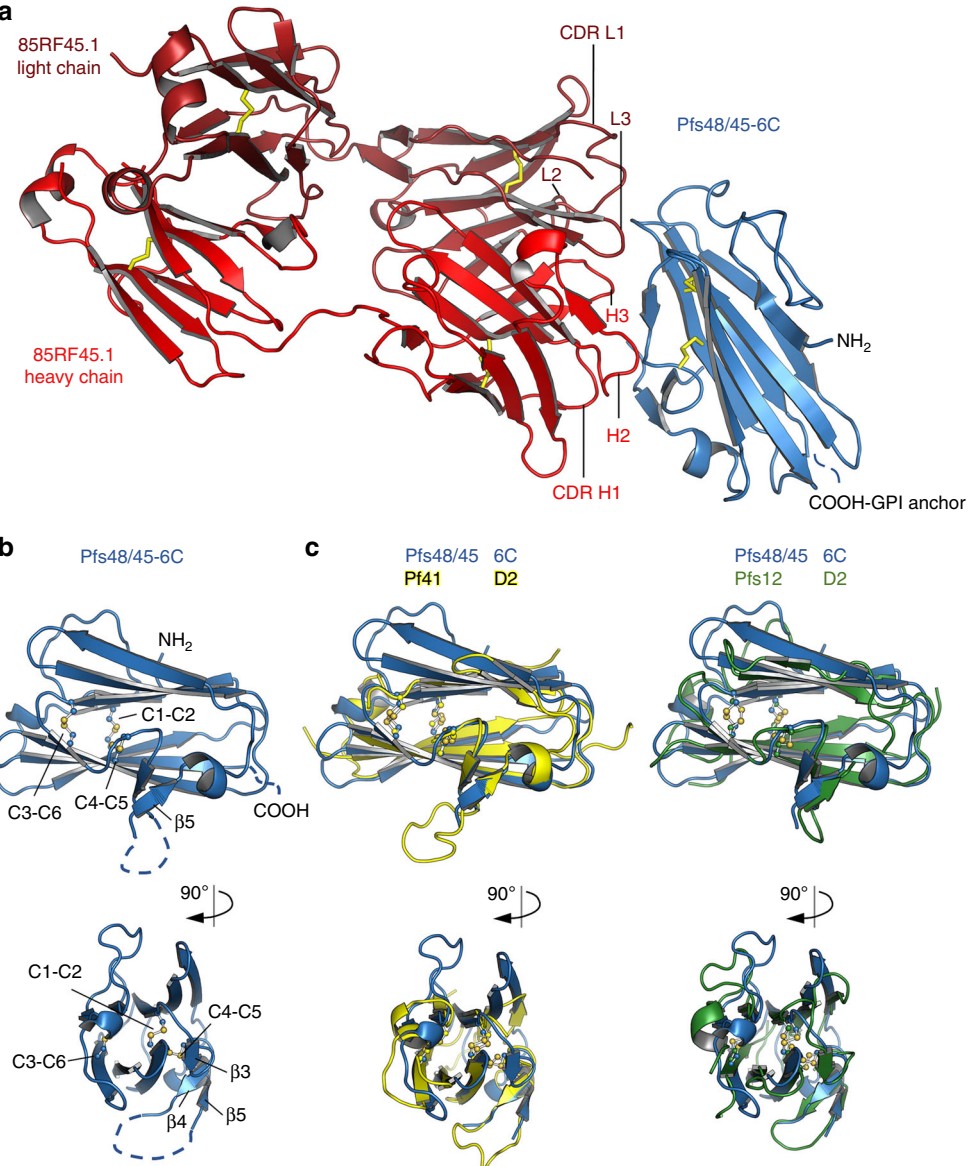

**Fig. 4** The structure of Pfs48/45-6C bound to a Fab fragment of the transmission-blocking mAb 85RF45.1. **a** The structure of Pfs48/45 (blue) bound to 85RF45.1 Fab fragment (red). The light (dark red) and heavy chains (light red) are indicated. Disulphide bonds are shown in stick representation. **b** Disulphide bond pattern within Pfs48/45-6C, numbered as in Fig. 1A. A disordered loop connecting β-sheet 4 and 5 is indicated by dashed lines. **c** Overlay of Pfs48/45-6C with the C-terminal 6-cys domains of Pf41 (PDB code 4YS4) and Pf12 (PDB code 2YMO). Pfs48/45$^{6C}$ and Pfs41$^{D2}$ were structurally aligned over 62 Cα atoms, Pfs48/45$^{6C}$ and Pfs12$^{D2}$ were aligned over 60 Cα atoms

C-terminal 6-cys domain, but also its central domain, suggesting that both domains should be explored as vaccine targets. This is consistent with previous studies which have analysed the antibodies generated by immunisation of rodents with either gametocytes [13] or individual domains of Pfs48/45 fused to large carrier proteins[28]. However, we find that, in addition to antibodies which reduce transmission in an SMFA, our immunisation experiment generates a larger number of antibodies with no transmission-blocking potential. Some of these bind to the C-terminal 6-cys domain, which is also the target of the best transmission-blocking antibody reported to date, 85RF45.1 (Fig. 2d)[6]. This indicates that simply including the C-terminal domain of Pfs48/45 in a vaccine will not selectively induce transmission-blocking antibodies, but might also raise ineffective antibodies. Designing an effective

vaccine candidate, that specifically elicits transmission-blocking activity, therefore requires a more focused approach and a better understanding of the spatial organisation of Pfs48/45 domains and its key epitopes.

For this, we determined the crystal structure of the most effective transmission-blocking antibody, 85RF45.1, bound to the C-terminal 6-cys domain of Pfs48/45. This structure reveals an interaction predominantly driven by electrostatics on a surface which is conserved in parasite genomes throughout the globe (Fig. 5). The Pfs48/45 binding site lies on the opposite face of the 6-cys domain to the GPI attachment site, through which Pfs48/45 is associated with the gamete membrane (Fig. 4 and Supplementary Figure 6). Parts of remaining surfaces of Pfs48/45 will associate with the N-terminal and central domains, which emerge

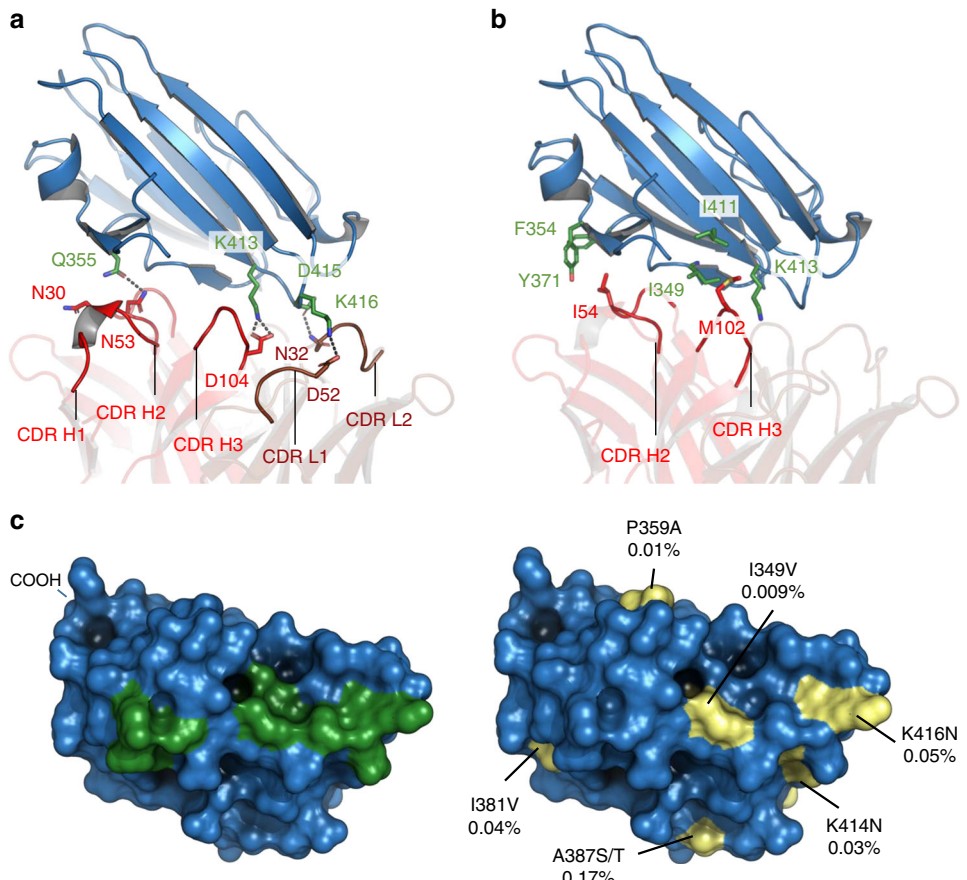

**Fig. 5** Determinants and conservation of the epitope targeted by 85RF45.1. **a**, **b** Direct interactions between the CDR loops of 85RF45.1 and charged residues (**a**) or hydrophobic patches (**b**) in Pfs48/45-6C. Heavy and light chain CDRs are coloured as in Fig. 4. Residues that directly interact are shown as stick representation (green for Pfs4845 residues, red and dark red for 85RF45.1 residues). Hydrogen bonds are indicated by dashed grey lines. **c** Surface footprint of 85RF45.1 on Pfs48/45-6C, represented by directly interacting residues, coloured in green. Residues that vary among >2400 Pfs48/45 sequences are shown in yellow and the observed substitutions as well as the overall frequency of substitutions are indicated

at the membrane distal side of the C-terminal domain. However, future structural studies will be needed to understand the context of the 85RF45.1-binding site relative to these two domains, and to understand the effect that the rest of the Pfs48/45 protein has on the exposure of this important epitope as well as revealing the epitopes for other antibodies with transmission-blocking potential. Future studies will also be needed to understand why 85RF45.1 is the most effective transmission-blocking antibody identified to date. In this work, we find that this is not due to improved affinity, as other antibodies, such as 32F3 have a similar affinity for Pfs48/45. It is instead likely that the high efficacy of 85RF45.1 is due to its ability to block the normal function of Pfs48/45, or its binding partners. As the molecular basis for the role of Pfs48/45 in gamete fusion is unknown, this will require further experimentation.

The structural studies presented here provide important insight into how to further develop Pfs48/45 as a vaccine candidate. First, small immunogens such as 6C are often presented on scaffolds, such as virus-like particles to boost their immunogenicity. Knowledge of the location of the 85RF45.1 epitope will guide such experiments, revealing the orientation in which this domain should be attached to such particles to elicit appropriate antibodies and prevent presentation of epitopes for non-transmission-blocking antibodies. Second, these findings pave the way for epitope grafting approaches in which the determinants from the 85RF45.1 epitope are grafted onto smaller

scaffolds to specifically elicit protective antibodies. Such structure-based approaches will now guide the development of more advanced immunogens for truly effective transmission-blocking vaccine components.

## Methods

**Expression and purification of Pfs48/45.** The Pfs48/45 sequence (PlasmoDB: PF3D7_1346700, residues 27–427) was codon optimised for expression in *Drosophila melanogaster* (GeneArt Life Technologies), the signal peptide was replaced with the *Drosophila* BiP signal peptide and the GPI-anchor sequence was replaced with the four amino acids EPEA (C-Tag). No further changes were made to the sequence and all N-glycosylation sites were left intact. The sequence was subcloned into the *Drosophila* S2 expression vector pExpres2.1 (ExpreS²ion Biotechnologies). Polyclonal Drosophila S2 stable cell lines were generated by non-viral transfection into ExpreS² Drosophila S² cells (Expres²ions Biotechnologies) and recombinant Pfs48/45 purified by buffer exchanging the cell culture supernatant using a Tangential Flow Filtration system with a Pellicon 3 Ultracel 10 kDa membrane (Merck Millipore, UK). The concentrated supernatant was then loaded onto Capture-Select™ C-tag affinity column (Thermo Fisher Scientific) equilibrated in Tris-buffered saline and bound proteins were eluted with 20 mM Tris–HCl, 2 M MgCl₂, pH 7.4. Fractions containing Psf48/45 were then pooled, concentrated and subjected to size-exclusion chromatography using a Superdex 200 16/60 PG column (GE Healthcare)[33]. 10C and 6C truncation variants were cloned, expressed and purified as described above. 10C comprises amino acids 159–428 while 6C comprises residues 291–428.

**Western blot and Coomassie.** Polyacrylamide gel electrophoresis was performed on pre-cast NuPAGE™ 4–12% Bis-Tris Midi Protein Gels (Invitrogen) polyacrylamide gels. Protein samples analysed under reducing conditions were incubated with 250 mM DTT before loading onto the gel. Gels were electrophoresed in

NuPAGE™ MES SDS Running Buffer (Invitrogen). Total protein in a gel was visualised by Coomassie staining (Quick Coomassie, Generon). Proteins were blotted on nitrocellulose membranes using the BioRad TransBlot Turbo Transfer System (BioRad). Different mouse and rat antibodies at a concentration of 1 μg/ml were bound to the membrane using the iBind Flex western system (Invitrogen), alkaline phosphatase-conjugated goat anti-mouse IgG (Sigma, A8438, 1:1000) or alkaline phosphatase-conjugated goat anti-rat IgG (Sigma, A8438, 1:1000) were used for detection. Western blots were developed using SIGMAFAST™ BCIP®/NBT (Sigma).

**Immunisation of mice with Pfs48/45**. All animal experiments and procedures were performed according to the UK Animals (Scientific Procedures) Act Project Licence (30/2889) and approved by the Oxford University Local Ethical Review Body. To assess immunogenicity of Pfs48/45-FL, six 6–8-week-old female CD1 mice (Envigo, UK) per group, housed in specific pathogen-free environments, were immunised with either 5 μg Pfs48/45-FL or 5 μg OVA three times with a 3-week interval. Group sizes between 6 and 10 animals were chosen as these numbers had been shown to allow for the detection of meaningful differences in immunogenicity in previous similar experiments. Immunisations were administered intramuscular in alternating hind legs in a total volume of 50 μl containing 25 μl AddaVax (InvivoGen). Prior to the second and third immunisation, serum samples were collected for analysis by ELISA. Three weeks after the third immunisations, mice were exsanguinated and serum collected for analysis by ELISA and SMFA. To obtain B cells for the generation of hybridoma cells, a 6–8-week-old female BALB/C mouse (Envigo) was immunised as above with 10 μg Pfs48/45-FL three times with a 3-week interval. Three weeks after the third immunisation the mouse received 10 μg Pfs48/45 intravenously to focus antigen-specific B cells in the spleen. Three days after the intravenous injection the mouse was killed and the spleen was collected.

**Generation of monoclonal antibodies and Fab fragments**. Hybridoma cells were generated and cultured using the ClonaCell™-HY Hybridoma Kit according to the manufacturer's manual. In brief, the spleen of an immunised mouse was dis-aggregated into a single-cell suspension and the splenocytes fused with SP2/0 myeloma cells (ECACC 85072401) by addition of PEG. Hybridomas were grown for 10 days in methylcellulose containing semi-solid medium under HTA selection to allow formation of individual colonies. Colonies were selected and transferred to 96-well plates and supernatant was positively screened for binding of Pfs48/45-FL and negatively screened for binding of Pfs25-C-Tag, by total IgG ELISA. Hybridomas showing a positive response to Pfs48/45 over two consecutive screens, but no response to Pfs25-C-Tag, were single-cell sorted on a Moflo FACS sorter (Beckman Coulter) and ELISA-positive sister clones were selected for cryopreservation as well as for further experiments. The isotypes of the mAbs were determined using the Pierce Rapid Isotyping Kit Mouse according to the manufacturer's instructions. For production of monoclonal antibodies, hybridomas were grown in Celline CL 1000 two-compartment bioreactors (Integra). Antibodies were purified from the supernatant using 5 ml HiTrap® Protein G columns (GE Healthcare) on an ÄKTA pure FPLC system (GE Healthcare). Fab fragments used for crystallisation trials were prepared from full-length 85RF45.1 mAb using Ficin immobilised on agarose (Thermo Fisher) according to the manufacturer's instructions. After cleavage, Fabs were further purified by size-exclusion chromatography on a Superdex Increase 75 10/300 column (GE Healthcare).

**Indirect fluorescence assay**. Cultured *P. falciparum* NF54 strain gametes were activated in FCS at room temperature for 30 min. Gametes were then air dried onto glass slides coated with poly-L-lysine and fixed with 4% paraformaldehyde. Slides were blocked for 1 h in blocking buffer (3% BSA/PBS) followed by incubation with test mAbs or control monoclonal antibody 32F3 in 3% BSA/PBS. The Rh5-specific mAb 2AC7 and PBS were used as negative controls. All mAbs were used at a concentration of 25 μg/ml. Slides were incubated at room temperature in a wet chamber for 1 h. Slides were washed three times in PBS and then incubated with AlexaFluor 488-conjugated goat anti-mouse IgG (0.5 μg/ml, Life Technologies A11029) for 1 h. For the last 5 min DAPI was added to a final concentration of 5 μg/ml. Slides were washed six times, mounted with IMM (ibidi) and analysed by fluorescence microscopy on a DMI3000B microscope (Leica Microsystems).

**Enzyme-linked immunosorbent assay (ELISA)**. Hybridoma supernatants were assessed by total IgG ELISA, and mAbs as well as antibody responses in mouse sera were assessed by end point total IgG ELISA. ELISAs were carried out in Nunc-Immuno Maxisorp 96-well plates (Thermo Scientific) coated with 2 μg/ml of antigen in carbonate–bicarbonate coating buffer (Sigma) overnight at 4 °C. Plates were washed with PBS-Tween and blocked with 10% Casein Block (Thermo Scientific). Hybridoma supernatants were added to the wells undiluted, while mAbs were diluted to a starting concentration of 5 μg/ml and sera were diluted to a starting concentration between 1:100 and 1:3000. Diluted samples for end point ELISAs were added to the top row of the plate in duplicate, and serially diluted threefold down the plate. Plates were incubated for 2 h at room temperature and then washed as before. Goat anti-mouse whole IgG conjugated to alkaline phosphatase (Sigma) was added for 1 h at room temperature. Following a final wash,

plates were developed by adding *p*-nitrophenylphosphate (Sigma) at 1 mg/ml in diethanolamine buffer (Sigma) and $OD_{405}$ was read on a microtitre plate reader (Tecan). End point titres were taken as the *x*-axis intercept of the dilution curve at an absorbance value of the background plus three standard deviations or a minimum of 0.25. If a 0.5 μg/ml dilution of an mAb or a 1:100 dilution of a serum sample did not develop a signal above background, the IgG titre in this sample was considered to be 0. To make results comparable, 32F3 was included on each plate as an internal control and all plates were developed until 32F3 reached an end point titre between 120,000 and 140,000.

**Cross-competition ELISA**. Monoclonal antibodies were binned into competition groups by cross-competition ELISA. Cross-competition ELISAs were carried out in Nunc-Immuno Maxisorp 96-well plates (Thermo Scientific) coated with 2 μg/ml of Pfs48/45-FL in carbonate–bicarbonate coating buffer (Sigma) overnight at 4 °C. Plates were washed with PBS-Tween and blocked with 10% Casein Block (Thermo Scientific). Blocking antibodies were added to the wells at a concentration of 25 μg/ml in triplicates. Fifteen minutes later, biotinylated antibody was added to the wells at a concentration of 2.5 μg/ml. Plates were incubated for 2 h at room temperature and then washed as before. Streptavidin conjugated to alkaline phosphatase (Mabtech 3310–10) was added for 1 h at room temperature to detect bound biotinylated mAbs. Following a final wash, plates were developed by adding *p*-nitrophenylphosphate (Sigma) at 1 mg/ml in diethanolamine buffer (Sigma) and $OD_{405}$ was read on a microtitre plate reader (Tecan). Competition was assessed as the blocking of a blocking antibody to reduce binding of a biotinylated antibody. The same non-biotinylated-blocking mAb was used as a positive control for competition for each biotinylated mAb, with the Rh5 mAb 2AC7 used as a negative control for competition. Biotinylated mAb added to wells without any blocking antibody was used as a positive control for detection of biotinylated mAbs. Biotinylated 2AC7 added to wells without any blocking antibody was used as the negative control.

**Surface plasmon resonance (SPR)**. All SPR experiments were carried out using a Biacore T200 instrument (GE Healthcare). To analyse binding of Pfs48/45-FL, Pfs48/45-10C and Pfs48/45-6C constructs to different mAbs, the Pfs48/45 constructs were buffer-exchanged into 20 mM HEPES pH 7.2, 300 mM NaCl, 0.05% Tween-20 and diluted to a concentration of 0.5 μM. The individual mAbs were then immobilised on a CM5-chip (GE Healthcare) pre-coupled to Protein A/G (Thermo Fisher) and the different Pfs48/45 constructs were then individually injected over the chip surface at a concentration of 0.5 μM and a flow rate of 30 μl/min, with 240 s association time and 240 s dissociation time. After each injection, the chip surface was regenerated with 10 mM glycine, pH 2.0 for 120 s at 10 μl/min, followed by a regeneration period of 180 s. All SPR data were analysed using the BIAevaluation software 2.0.3 (GE Healthcare).

**Standard membrane-feeding assay (SMFA)**. SMFA was performed to assess the ability of polyclonal and monoclonal antibodies to block the development of *P. falciparum* strain NF54 oocysts in the mosquito midgut[34]. For this, stage V gametocytes from a mature gametocyte culture were mixed with normal human serum and normal red blood cells to make a feeding mixture with 0.15–0.2% stage V gametocytemia. Unless stated, SMFAs were conducted in the presence of active human complement. Purified IgG or mAbs were added to these at the specified concentrations and then fed to 3–6-day-old starved female *A. stephensi* (SDA 500, NIAID) via a parafilm® membrane. The mosquitoes were maintained for 8 days and then dissected to count the number of oocysts per midgut in 20 mosquitoes. Percent reduction in infection intensity was calculated relative to the respective control IgG tested in the same assay.

**Statistical analysis**. Comparison of antibody titres within the same group but at different time points were performed using Wilcoxon-matched pairs signed-rank test. Antibody titres between different groups were compared by Mann–Whitney test.

TRA was calculated from SMFA data, as 100 × (mean number of oocysts in test/mean number of oocysts in control) and 95% confidence intervals (95% CIs) of % inhibition in oocyst density from a single- or multiple-feeding experiments for each test antibody at each concentration were calculated using a zero-inflation negative binomial model[34]. Statistical tests were performed using Prism 6 (GraphPad Software Inc, USA) or JMP11 (SAS Institute Inc, USA). *P* values <0.05 were considered significant.

**Circular dichroism (CD) spectroscopy**. Far-UV CD spectroscopy experiments were conducted on a J-815 Spectropolarimeter attached to a Peltier temperature control unit. For the measurements, the samples were dialysed against 100 mM sodium phosphate buffer, 150 mM NaF, pH 7.2 and diluted to 0.4 mg/ml. CD spectra recorded at 20 °C between wavelengths of 195 and 260 nm using a cell with a 1 mM path.

**Crystallisation**. A complex of Pfs48/45-6C with a Fab fragment from 85RF45.1 mAb was generated by mixing purified Pfs48/45-6C with a 1.5 molar excess of

purified Fab, followed by size-exclusion chromatography on two Superdex Increase 200 10/300 columns (GE Healthcare) in series into 10 mM HEPES, 150 mM NaCl, pH 7.2. Peak fractions containing pure complex were then concentrated and used to set up vapour diffusion crystallisation trials in sitting drops by mixing 100 nl of protein complex with 100 nl of well solutions. Initial crystals for Pfs48/45-6C–85RF45.1 Fab grew at 277 K at a concentration of 11.7 mg/ml as small stacks of plates in conditions from the JCSG + screen (Molecular Dimensions) containing 0.2 M ammonium dihydrogen phosphate, 0.1 M Tris pH 8.5 and 50% (v/v) 2-methyl-2,4-pentanediol. Crystals were further optimised by microseeding into 0.1 M Tris pH 8.0 and 50% (v/v) 2-methyl-2,4-pentanediol which yielded slightly larger, individual plates suitable for diffraction experiments. These crystals were harvested, transferred into well solution containing 0.1 M Tris pH 8.0, 50% (v/v) 2-methyl-2,4-pentanediol and 25% glycerol and flash frozen in liquid nitrogen for cryoprotection.

**Data collection, phasing and refinement**. Data were collected at the IO3 beamline (Diamond Light Source, UK) with a Pilatus3 6M detector (Dectris, Baden-Daettwil, Switzerland) and X-rays at a wavelength of 0.98 Å. Data were processed with the CCP4i2 programme suite[35] using the Xia2/DIALS pipeline[36] for indexing and scaling. The structure was solved by molecular replacement with Phaser-MR[37], using a poly-alanine model of the most closely related Fab fragment (PDB code 4k2u [https://www.rcsb.org/structure/4K2U][29]) with trimmed loops as search model. Molecular replacement found two copies of the Pfs48/45-6C–85RF45.1 Fab complex in the asymmetric unit. After one initial round of rigid-body refinement using the PHENIX programme suite[38], density for Pfs48/45-6C was clearly visible, and both Pfs48/45-6C and the remaining parts of the 85RF45.1 Fab fragment were built by iterative cycles of model building in Coot[39] and refinement in Buster[40]. Both copies of Pfs48/45 and the variable domains of both copies of 85RF45.1 were well defined. The constant domains of 85RF45.1 were well defined in chains B and C, but density was poor in chains E and F. For this reason, the constant domains of 85RF45.1 were built in copies B and C and used as a molecular replacement search model in phaser to identify the correct location for the equivalent domains in chains E and F. The structure was refined to give final Ramachandran statistics of 93.4% residues in the favoured regions, 6.6% in the allowed regions and no residues in the disallowed regions. The coordinate and structure factor data are deposited in the protein data bank (PDB) under the accession code 6H5N. All figures showing structures were prepared with PyMol (Schroedinger LLC).

## Data availability

Data for the structure reported here have been deposited in the PDB under the accession code 6H5N. Additional data supporting the findings reported in this manuscript are available from the corresponding authors on request.

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

## Acknowledgements

This work was funded by a Medical Research Council project grant (MR/R001138/1) to S.B. and M.K.H. M.K.H. is a Wellcome Investigator. F.B. is a recipient of a Wellcome-Trust 4-years PhD grant (102051/Z/13/Z). The SMFA activity determination was supported by the intramural programme of the National Institute of Allergy and Infectious Disease/NIH and by PATH's Malaria Vaccines Initiative. We would like to thank Katrina Teelen for production of monoclonal antibodies 32F3, 85RF45.1, 85RF45.3 and 85RF45.5. This publication uses data generated by the Pf3k project (www.malariagen.net/pf3k) and in the Pf3K project (2015): pilot data release 3. The authors would specifically like to thank Dr Richard Pearson for help with the interpretation of the Pf3K data.

## Author contributions

F.L., F.B., S.B. and M.K.H. conceived and planned the study and wrote the manuscript. M.M.S., T.J. and W.A.d.J. generated the cell line for the production of Pfs48/45-FL. R.D. produced Pfs48/45-FL and characterised the polyclonal immune response. D.M. prepared western blots and Coomassie gels. K.M. and C.A.L. performed SMFAs. F.B. generated monoclonal antibodies against Pfs48/45-FL and characterised the mAbs by ELISA, IFA and dot blot. R.W.S. and M.M.J. provided 32F3, 85RF45.1, 85RF45.3 and 85RF45.5. M.M.J. prepared gamete slides for IFA. R.D. generated cell lines for the production of Pfs48/45-6C and 10C. R.D. and D.M. expressed and purified Pfs48/45-6C and 10C. F.L. prepared Fab fragments, purified and crystallised complexes and collected X-ray diffraction data. F.L. and M.K.H. built, refined and analysed the structure. F.L. performed and analysed SPR and CD experiments. FB analysed Pf3k data to determine the sequence conservation of Pfs48/45 epitopes. All authors read and commented on the manuscript.

## Additional information

**Competing interests:** M.M.S., T.J. and W.A.d.J. are employees of and W.A.d.J. is a shareholder in ExpreS$^2$ion Biotechnologies, which has developed and is marketing the ExpreS$^2$ cell expression platform. The remaining authors declare no competing interests.

