## [Peer Review File · Nature Communications]

Reviewers' comments:

Reviewer #1 (Remarks to the Author):

Malaria transmission-blocking vaccines (TBVs) have been regarded as one of the essential strategies towards malaria elimination. Pfs48/45 was considered to be a promising TBV candidate for a long time. In this study, the Authors successfully produced full-length Pfs48/45 for the first time and raised a panel of mAbs. Then they mapped the binding regions of these antibodies on Pfs48/45 and correlated the location of their epitopes with their transmission-blocking activity. More importantly, they solved the structure of the Pfs48/45 6C which contains an epitope of the most potent transmission-blocking antibody 45.1. I think the results of this study will provide the TBV community with very important information for developing effective TBVs based on Pfs48/45. All the works were carefully designed, clearly presented, and the manuscript is well written. I have the following comments for the improvement of this manuscript.

Comments:

Major

- 1) This is the first success of the properly folded full-length Pfs48/45 expression. So the detail information how they achieved the success is definitely required, especially how they modify the glycosylation sites.
- 2) They produced a panel of mAbs with different epitopes, affinity, and efficacy as summarized in Fig 3B. These data are very important to understand how these mAbs block or not block the biological activity of Pfs48/45. I strongly recommend the Authors to add one additional discussion why 45.1 was the strongest among their mAbs tested.
- 3) Related to my comment 2) above: Page 10, Lines 10-13
"This indicates that the efficient and specific elicitation of transmission-blocking antibodies will not be brought about simply by inclusion of an individual domain of Pfs48/45 in a vaccine, but requires a more focused approach and a better understanding of the spatial organization of Pfs48/45 and its key epitopes."
Please clarify this sentence.

Minor

- 1) mAb 45.1
Please change this to mAb "85RF45.1" throughout this manuscript as described in the original paper (Ref #13).
- 2) SMFA and isotype of the mAbs
Does this SMFA method contain complement? If complement is present in the SMFA, are there any correlations between the TRA and isotype of each mAb?
- 3) Page 14, Line 2 from the bottom
"in 35." Please remove "in".

Reviewer #2 (Remarks to the Author):

This paper focuses on Pfs48/45, which is a promising target for a transmission blocking vaccine against malaria. The authors express the full length protein, raise monoclonal antibodies to different domains and mapped the epitopes and describe their transmission blocking activity. They also obtain a structure of the C-terminal domain bound to a previously known transmission blocking antibody.

The study is well written and should be of interest to the malaria vaccine field.

There are changes that the authors should consider before submission.

The authors group the antibodies in competition groups although epitopes have been defined previously against that protein. It might be clearer to relate the competition group to the previously described epitopes. In Fig 2B, it might help to clarify which domains the antibodies bind to.

Is there an explanation for some of the antibodies binding more "weakly" to 6C vs 10C or FL (Fig 3A)?

The antibodies seem to bind with different strength to the various proteins and this could be seen better if the same y axis was used. Maybe a comment on why that is could be added.

It will be interesting to know if the authors have tried co-crystallization of the antibody 45.1 with the 10C protein or if they have tried co-crystallization of the new antibodies they describe in complex with their proteins.

Minor: Figure 3B seems backward with the C-terminus to the left and the N-terminus to the right.

Fig 4A, it looks like the disulfide bond shown as stick is not "attached" to the cartoon representation.

Reviewer #3 (Remarks to the Author):

This is the first report of the crystal structure of a Pfs48/45-specific malaria transmission blocking monoclonal antibody and a recombinant Pfs48/45 protein. It extends our understanding of the structure of the 6-cys motif domain in this important malaria vaccine candidate and provides insights into the design of future recombinant vaccine candidates that are directed against a specific transmission blocking epitope. The importance of targeting a key epitope is demonstrated by the number of Pfs48/45 specific mAb they generated against different forms of recombinant Pfs48/45, including the same section used for the crystal, that did not significantly reduce transmission. Using a drosophila cell expression system they also for the first time successfully produced full-length, recombinant

Pfs48/45 without a fusion protein and showed that it induced transmission-blocking Ig in outbred mice (CD-1). This is a promising vaccine candidate, depending on the economics of scaling up production. I only have a few suggestions,

1) One is to add information on the glycosylation state of the recombinant proteins, including whether the N-linked glycosylation sites have been modified.

2) The other is to evaluate the ability of their new monoclonals to bind directly to intact gametes, which would indicate if the epitopes are actually exposed on the surface. If some are not exposed it would be interesting to test the Pfs230 disruptant parasites to see if binding is enhanced.

Introduction

Pg 3)

Second line of the first full paragraph, In reference 5 it is my understanding that the lack of gamete fusion was only formally been shown in *P. berghei* Pfs48/45 KO mice.

Second line of the first full paragraph, anti-Pfs48/45 antibodies block oocyst formation. If one of the references listed demonstrates that antibodies block fusion, please specifically indicate which reference tested gamete fusion directly in *P. falciparum*.

Results

Pg 7) Please include whether attempts were made to crystalize the full length and 10C recombinant proteins and whether they did or did not form usable crystals

Discussion:

In addition to adding information about the potential effect of glycosylation, please add a brief comment about the potential differences between the efficacy of the polyclonal serum and individual mAb, including the possible influence of using CD1 vs BALB/C mice. Was polyclonal serum from the BALB/C mice testing for transmission-blocking activity?

It would be good to include a brief speculation about the location of the interaction of Pfs48/45 with Pfs230, especially if the N-terminal region mAb do not recognize the surface of intact gametes.

Figures:

Fig 3B) I would flip the orientation of the schematic so the C-terminus is at the right and the N-terminus is at the left to be consistent with the standard depiction of protein sequences.

Fig S2 and S4C: The patterns would be more clear if the Ig were ordered by cluster analysis or at least by group, not numerically.

Fig S7: It would be good to indicate the location of the epitope for 45.1 on the 6C bar at the top of the figure.

Table S2 is a nice summary of the specific interactions.

Reviewer 1:

1) This is the first success of the properly folded full-length Pfs48/45 expression. So the detail information how they achieved the success is definitely required especially how they modify the glycosylation sites.

The reviewer asks about the glycosylation state of Pfs48/45. We made no changes to the construct which alter the native protein sequence and therefore the glycosylation sites were left intact. We have now stated this in the results section on page 5 line 1 and in the methods section on page 12 line 12-13.

2) They produced a panel of mAb with different epitopes, affinity, and efficacy as summarized in Fig 3B . These data are very important to understand how these mAbs block or not block the biological activity of the Pfs48/45. I strongly recommend the Authors to add one additional discussion why 45.1 was strongest among their mAbs tested.

The reviewer asks us to explain why 45.1 is the most effective transmission blocking antibody. Sadly, we are not in a position to come to a conclusive response to this question. We are, however, able to rule out some possibilities. In SPR experiments, we observe similar overall binding strengths of the different inhibitory antibodies. This is also true for blocking antibodies (compare, for example, 85RF45.1 and 32F3 in Figure 3A) that target the C-terminal 6-cys domain, indicating that the reason for the effectiveness of 85RF45.1 is not a result of increased binding strength. We also observe that Pfs48/45-6C is the target of both inhibitory and non-inhibitory antibodies, leading us to speculate that the reason for the increased blocking activity of 85RF45.1 is due to the specific localisation of its epitope on Pfs48/45 rather than its epitope being within a particular domain. Instead it seems likely that 85RF45.1 is blocking a functional region of Pfs48/45. However, as the molecular basis for the function of Pfs48/45 is not yet understood, we are not currently in a position to comment on this. We have added a brief section to the discussion in page 11 lines 19-25 to make this point.

3) Related to my comment 2) above: Page 10, Lines 10-13

“This indicates that the efficient and specific elicitation of transmission blocking antibodies will not be brought about simply by inclusion of an individual domain of Pfs48/45 in a vaccine, but requires a more focused approach and a better understanding of the spatial organisation of Pfs48/45 and its key epitopes.”

Please clarify this sentence.

We thank the reviewer for highlighting a confusing section in the text. We have re-written this section, on page 11 lines 2-6 and hope that it is clearer.

1) mAb 45.1

Please change this to mAb “85RF45.1” throughout this manuscript as described in the original paper (Ref #13).

We have changed 45.1 to 85RF45.1 throughout the text as requested.

2) SMFA and isotype of the mAbs

Does this SMFA method contained complement? If complement is present in the SMFA, are there any correlation between the TRA and isotype of each mAb?

The reviewer asks for clarification of the method used for the SMFA studies. SMFAs were conducted in the presence of active human complement, as now mentioned in the methods on page 16 line 18-19. However, we observe similar transmission reducing activity from polyclonal anti-Pfs48/45 serum when SMFA was performed in the absence of complement. This data is now included in Figure S1C and D.

The reviewer also asked about the isotypes of the mAbs in the newly generated panel. We have now determined the isotypes using the method described on page 13 lines 31-30 and include the data in Figure 2A. No correlation could be detected between isotype and transmission reducing activity. This point is now made on page 6 lines 25-26.

These data suggest that, unlike in the case of anti-Pfs230 IgG, complement does not play a major role in anti-Pfs48/45 IgG mediated activity.

3) Page 14, Line 2 from the bottom

“in 35.” Please remove “in”.

This correction has been made.

Reviewer 2:

The authors group the antibodies in competition groups although epitopes have been defined previously against that protein. It might be clearer to relate the competition group to the previously described epitopes. In Fig 2B, it might help to clarify which domains the antibodies bind to.

As suggested, to help readers to relate our findings to previous studies, we have indicated the location of previously defined epitopes I-V on our schematic in Figure 3B. As this schematic also includes what we know about the binding sites for the new mAbs, we hope that this provides the required clarity.

Is there an explanation for some of the antibodies binding more "weakly" to 6C vs 10C or FL (Fig 3A)?

The antibodies seem to bind with different strength to the various proteins and this could be seen better if the same y axis was used. Maybe a comment on why that is could be added.

The reviewer comments on the SPR data presented in Figure 3A, mostly focusing on the relative affinities of the mAbs. Our major aim in this figure was not to measure the affinities of mAbs, for which we would have conducted a more extensive analysis at different concentrations, but to map which mAbs bind to which domains of Pfs48/45. Our decision to show these curves on different y axis scales was made with the aim of more clearly highlighting these differences, rather than the differences in affinity, and we believe that these conclusions will be made more clearly to readers if we keep this format.

In terms of what we can learn about antibody affinities, in the majority of cases the curves show the expected magnitudes, with the more massive FL generating a greater response than the smaller 10C, and 10C generating a higher response than the even smaller 6C. For three of the antibodies we observe higher binding of the Pfs48/45-10C construct than the full-length Pfs48/45. We believe that the reason for this phenomenon is better overall accessibility of the epitope in the 10C construct as compared to full-length Pfs48/45. We have added to the text on page 7 lines 21-28.

It will be interesting to know if the authors have tried co-crystallization of the antibody 45.1 with the 10C protein or if they have tried co-crystallization of the new antibodies they describe in complex with their proteins.

The reviewer asks if we have tried to crystallise other protein complexes. These experiments are ongoing but have not reached the stage where we have other completed structures to report.

Minor: Figure 3B seems backward with the C-terminus to the left and the N-terminus to the right.

Fig 4A, it looks like the disulfide bond shown as stick is not "attached" to the cartoon representation.

The reviewer suggests that we alter the orientation of Figure 3B such that the N-terminal epitope is at the left of the page. We have made this change.

Reviewer 3:

1) One is to add information on the glycosylation state of the recombinant proteins, including whether the N-linked glycosylation sites have been modified.

Please see our response to reviewer 1 point 1

2) The other is to evaluate the ability of their new monoclonals to bind directly to intact gametes, which would indicate if the epitopes are actually exposed on the surface. If some

are not exposed it would be interesting to test the Pfs230 disruptant parasites to see if binding is enhanced.

The reviewer asked us to assess the binding of our new mAbs to gametes. The outcome of an immunofluorescence experiment has now been included in a new Supplementary Figure 3 and is described on page 6 lines 16-19. All but three of the mAbs label the gametes. The three mAbs that do not show efficient labelling of gametes are those shown by SPR to have the weakest affinities for Pfs48/45 and the highest dissociation rates. Their inability to bind to gametes could therefore be due to these unfavourable binding characteristics rather than to the epitopes not being exposed on the surface. This interpretation is supported by the very weak fluorescence signal detected with 7A7, which suggests weak binding rather than occlusion of the epitope. We now make this point on page 7 lines 24-28.

Second line of the first full paragraph, In reference 5 it is my understanding that the lack of gamete fusion was only formally been shown in *P. berghei* Pbs48/45 KO mice.

The reviewer helpfully points out that the lack of attachment and fusion of male P48/45 knockout gametes to female gametes was only shown for *P. berghei*. We have therefore re-written this sentence, in page 3 lines 11-15, to clarify this and have added the information that ookinete formation was inhibited in Pfs48/45 gametes, leading to severely reduced transmission rates to the mosquito host.

Second line of the first full paragraph, anti-Pfs48/45 antibodies block oocyst formation. If one of the references listed demonstrates that antibodies block fusion, please specifically indicate which reference tested gamete fusion directly in *P. falciparum*.

The reviewer asked for clarification of our claim that Pfs48/45 targeted antibodies can block oocyst formation. While these studies do not directly test the fusion event, they do assess, through SMFA experiments, the presence of oocysts in the midgut of mosquitoes. We have checked the sentences and believe that we have expressed this correctly.

Pg 7) Please include whether attempts were made to crystalize the full length and 10C recombinant proteins and whether they did or did not form usable crystals

Please see our response to reviewer 2 point 3

In addition to adding information about the potential effect of glycosylation, please add a brief comment about the potential differences between the efficacy of the polyclonal serum and individual mAb, including the possible influence of using CD1 vs BALB/C mice. Was polyclonal serum from the BALB/C mice testing for transmission-blocking activity?

The reviewer asks about the decisions that we made in choosing which strains of mice to use for the different types of analysis. For the generation of polyclonal serum outbred (CD1) mice were used to maximise the breadth of the induced antibody response. In contrast, Balb/c mice were used to generate monoclonal antibodies as they have previously been used to produce highly transmission blocking monoclonal antibodies, including 32F3, which was used as a reference antibody in most of our assays. We did not test the transmission reducing activity of polyclonal anti-Pfs48/45 serum from Balb/c mice as we feel that the data collected from outbred mice is more appropriate for the analysis of polyclonal responses. We have now clarified these points, and the reasons for decisions made in the manuscript, with new text on page 5 lines 10-11 and page 5 lines 30-31.

It would be good to include a brief speculation about the location of the interaction of Pfs48/45 with Pfs230, especially if the N-terminal region mAb do not recognize the surface of intact gametes.

The reviewer suggests that we speculate about the interaction site of Pfs48/45 and Pfs230. However, we are not aware of any data about the molecular details of this interaction that would back up any such conclusions, especially in view of the ability of mAbs that bind to each of the three domains of Pfs48/45 to recognize gametes in an IFA experiment. We therefore do not feel comfortable to discuss this at the current time.

Fig 3B) I would flip the orientation of the schematic so the C-terminus is at the right and the N-terminus is at the left to be consistent with the standard depiction of protein sequences.

- 1) The reviewer suggests that we alter the orientation of Figure 3B such that the N-terminal epitope is at the left of the page. We have made this change.

Fig S2 and S4C: The patterns would be more clear if the Ig were ordered by cluster analysis or at least by group, not numerically.

- 2) Following the suggestion of the reviewer, we have rearranged Figures S2 and S4C so that the mAbs are organized according to their groups.

Fig S7: It would be good to indicate the location of the epitope for 45.1 on the 6C bar at the top of the figure.

- 3) In Figure S7 (Fig S8 in the final version of the manuscript), we have now indicated the location of the 45.1 epitope.

REVIEWERS' COMMENTS:

Reviewer #1 (Remarks to the Author):

The Authors responded appropriately to all the comments.

Reviewer #3 (Remarks to the Author):

The authors addressed most of the comments, but missed three references to the function of Pfs48/45 that need to be revised or a Pf reference added.

Pg 3: Line 12: Need a reference for impaired Pf ookinete formation. Would also need a reference if they switch to oocyst formation.

Pg 3: Line 16: Need a reference for blocking Pf gamete fusion, or switch to oocyst formation

Pg11: 24 Need a reference for anti-Pfs48/45 Ig blocking Pf gamete fusion, or switch to oocyst formation

In response to reviewer 3:

The authors addressed most of the comments, but missed three references to the function of Pfs48/45 that need to be revised or a Pf reference added.

We thank the reviewer for pointing out an inaccuracy in one of our statements and requesting a new reference and we have modified the manuscript accordingly.

Pg 3: Line 12: Need a reference for impaired Pf ookinete formation. Would also need a reference if they switch to oocyst formation.

We have added a reference (van Dijk et al) describing the impaired Pf ookinete formation to the manuscript.

Pg 3: Line 16: Need a reference for blocking Pf gamete fusion, or switch to oocyst formation

We thanks the reviewer for pointing this out; the references cited show that antibodies against Pfs48/45 very effectively block transmission and subsequent sexual development in mosquitos; Blocking Pf gamete fusion has formally not been shown in these references. We have therefore modified the sentence accordingly.

Pg11: 24 Need a reference for anti-Pfs48/45 Ig blocking Pf gamete fusion, or switch to oocyst formation

We couldn't locate this citation on Page 11 line 24, but where we found a statement to this effect (on page 3) we have re-written as above.